# Genomic Instability in Cerebrospinal Fluid Cell-Free DNA Predicts Poor Prognosis in Solid Tumor Patients with Meningeal Metastasis

**DOI:** 10.3390/cancers14205028

**Published:** 2022-10-14

**Authors:** Peng Wang, Qiaoling Zhang, Lei Han, Yanan Cheng, Zengfeng Sun, Qiang Yin, Zhen Zhang, Jinpu Yu

**Affiliations:** 1Department of Neuro-Oncology and Neurosurgery, Tianjin Medical University Cancer Institute and Hospital, National Clinical Research Center for Cancer, Key Laboratory of Cancer Prevention and Therapy, Tianjin 300060, China; 2Tianjin’s National Clinical Research Center for Cancer, Tianjin 300060, China; 3Cancer Molecular Diagnostics Core, Tianjin Medical University Cancer Institute and Hospital, National Clinical Research Center for Cancer, Key Laboratory of Cancer Immunology and Biotherapy, Tianjin 300060, China

**Keywords:** genomic instability, meningeal metastases, cerebrospinal fluid, plasma, cell-free DNA

## Abstract

**Simple Summary:**

We established a genomic instability score using unfiltered sequencing data from meningeal metastasis (MM) cell-free circulating tumor DNA (ctDNA) samples and found that substantial genomic instability (GI) was present in cerebrospinal fluid ctDNA rather than plasma ctDNA, implying that MM lesions have a significantly increased GI status compared to primary tumors or extracranial metastatic lesions, which may suggest tumor clonal evolution. We also found that high GI status was an independent poor prognostic factor in lung adenocarcinoma MM patients, including meningeal metastasis-free survival (MFS) and overall survival (OS). Considering that genomically unstable tumors are more sensitive to PARP inhibitors, targeting GI alone or in combination with conventional therapy may be a promising treatment strategy for solid tumor patients with MM.

**Abstract:**

Genomic instability (GI), which leads to the accumulation of DNA loss, gain, and rearrangement, is a hallmark of many cancers such as lung cancer, breast cancer, and colon cancer. However, the clinical significance of GI has not been systematically studied in the meningeal metastasis (MM) of solid tumors. Here, we collected both cerebrospinal fluid (CSF) and plasma samples from 56 solid tumor MM patients and isolated cell-free ctDNA to investigate the GI status using a next-generation sequencing-based comprehensive genomic profiling of 543 cancer-related genes. According to the unfiltered heterozygous mutation data-derived GI score, we found that 37 (66.1%) cases of CSF and 3 cases (6%) of plasma had a high GI status, which was further validated by low-depth whole-genome sequencing analysis. It is demonstrated that a high GI status in CSF was associated with poor prognosis, high intracranial pressure, and low Karnofsky performance status scores. More notably, a high GI status was an independent poor prognostic factor of poor MM-free survival and overall survival in lung adenocarcinoma MM patients. Furthermore, high occurrences of the co-mutation of *TP53/EGFR, TP53/RB1, TP53/ERBB2*, and *TP53/KMT2C* were found in MM patients with a high GI status. In summary, the GI status in CSF ctDNA might be a valuable prognostic indicator in solid tumor patients with MM.

## 1. Introduction

Meningeal metastasis (MM), also known as neoplastic meningitis or meningeal carcinomatosis, is the diffuse dissemination of tumor cells into leptomeninges and the cerebrospinal fluid (CSF) [1]. It occurs in cancers that are most likely to spread to the central nervous system, including breast cancer, lung cancer, colorectal cancer and melanoma [2]. There is no cure for MM, and treatment is mainly aimed at prolonging survival and stabilizing neurological symptoms. Approaches include systemic or intrathecal chemotherapy, targeted therapy, radiation therapy, and surgery [3]. In recent years, the use of EGFR-tyrosine kinase inhibitors such as erlotinib and osimertinib has markedly prolonged survival for patients with MM from EGFR mutation-positive NSCLC [4,5]. However, the aggressive nature and increasing prevalence of MM underscore the need to understand the underlying mechanisms of MM and to explore new therapeutic strategies.

Genomic instability (GI) has very complex effects on tumors, including the loss or amplification of driver genes, large fragment rearrangements, extrachromosomal DNA, micronucleus formation, and the activation of innate immune signaling, among others [6]. GI is associated with disease staging, metastasis, poor prognosis, and treatment resistance [7,8]. Studies have reported that the progression of many tumors (e.g., breast, ovarian, colorectal, etc.) can be driven by GI, allowing tumors to consistently generate new genetic variants that provide a selective growth advantage for tumor cells and ultimately lead to poor clinical outcomes and unfavorable objective responses to conventional treatment paradigms [9]. An analysis of the genomic alterations from unpaired primary and metastatic samples revealed that brain metastasis is associated with a higher level of chromosomal instability (a form of genome instability), as inferred by a higher fraction of altered genomes compared to primary tumors (lung adenocarcinoma (LUAD)) [10]. At the same time, GI is expected to be translated into therapeutic targets [11]. For example, the induction of genomic damage in genomically unstable tumors, such as breast cancer and ovarian cancer, holds promise for the application of synthetic lethal principles (e.g., poly (ADP-ribose) polymerase [PARP] inhibitors) to establish new chemotherapeutic regimens [12,13]. 

However, until our study last year, few studies specifically examining GI in patients with MM had been conducted [14]. Normally, the GI status is analyzed using genomic DNA samples from tumor tissues rather than cell-free circulating tumor DNA (ctDNA) extracted from plasma, though, in many cases, cfDNA has been used as a surrogate reflecting the genomic profile of solid tumors for concomitant diagnosis, minimal residual disease (MRD) surveillance, and the identification of drug resistance mechanisms [15,16,17]. Since MM tissues are rarely collected in clinics, the quantitative GI metrics and their clinical value in solid tumor patients with MM remain unmentioned. CSF is a circulating fluid unique to the central nervous system, filling the ventricles and spinal cord, that comes into direct contact with tumor cells at intracranial lesions and is expected to be a breakthrough point for the liquid biopsy of brain tumors [18,19,20]. In our previous study, we found that the GI status in the MM of multiple solid tumors could be analyzed using CSF cfDNA [14]. In the present study, we established a GI score relying on high-throughput genetic testing to describe the GI status in CSF cfDNA and assess its clinical significance as a prognostic indicator in solid tumor patients with MM.

## 2. Materials and Methods

### 2.1. Patient Information

We collected a total of 56 cerebrospinal fluid (CSF) samples and 50 paired plasma samples from patients with meningeal metastasis (MM) treated at the Tianjin Medical University Cancer Institute and Hospital (TJMUCH) from June 2012 to September 2018 (Table 1). The primary tumor types of a total of 56 MM patients included 45 cases of lung adenocarcinoma (LUAD), 7 cases of breast carcinoma (BRCA), 1 case of colon adenocarcinoma (COAD), 1 case of stomach adenocarcinoma (STAD), 1 case of small cell lung cancer (SCLC), and 1 case of lung squamous cell carcinoma (LUSC). Nine cases of primary tumor tissue samples were obtained, including six LUAD, one SCLC and two BRCA. All patients were diagnosed with MM since distinct cancer cells were detected in the CSF samples by professional pathologists. The project was approved by the Ethics Committee of Tianjin Medical University, and written informed consent was obtained from the patients. 

### 2.2. Plasma Cell-Free DNA Extraction

The supernatant plasma obtained from the centrifugation (1600× *g*, 10 min) of peripheral blood lymphocytes and plasma was transferred to a 2 mL centrifuge tube and further centrifuged at 16,000× *g* for 10 min. Plasma Cell-Free DNA (cfDNA) was isolated by the MagMAXTM Cell-Free DNA Isolation Kit (Life Technologies, Carlsbad, CA, USA). Following the protocol, DNA was extracted from peripheral blood lymphocytes using the Tiangen Whole Blood DNA kit (Tiangen, Beijing, China). DNA concentrations were quantified using the Qubit dsDNA HS Assay Kit or the Qubit dsDNA BR Assay Kit (Life Technologies, USA).

### 2.3. cfDNA Extraction from CSF

The supernatant obtained from CSF centrifugation (3000× *g*, 15 min) was transferred to a 5 mL centrifuge tube, followed by centrifugation at 16,000× *g* for 10 min. The CSF supernatant cfDNA and DNA were extracted from CSF using the MagMAXTM Cell-Free DNA Isolation Kit and Dynabeads Myone Silane Isolation Kit. DNA concentration was quantified by the Qubit dsDNA HS kit or the Qubit dsDNA BR kit.

### 2.4. Library Preparation and Next-Generation Sequencing

Genomic DNA was sheared into 150–200 bp fragments by a Covaris M220 focused sonicator (Covaris, Woburn, MA, USA), and then fragmented DNA libraries were created using the KAPA HTTP library preparation kit (Illumina Platform) (KAPA Biosystems, Woburn, MA, USA). DNA libraries were captured by a designed NimbleGen SeqCap EZ library (Roche, Madison, WI, USA), a panel containing, typically, 543 cancer-related genes, followed by Novaseq 6000 paired-end sequencing. The DNA concentration, sequencing depth, and library input were 0.02–79.4 ng/µL, 471–9770, and 1–401 ng, respectively.

### 2.5. Low-Throughput Whole-Genome Sequencing

The starting amount of ctDNA for building the library was 10 ng, and the DNA was fragmented into an average size of 320 bp. Sequencing libraries purified by glass fiber membrane purification technology were massively sequenced in parallel by the Illumina NovaSeq 6000 platform and NovaSeq 6000 S4 kit v1.5 (300 runs).

### 2.6. Calculation of the GI Score

The theoretical value of allele frequency (AF) for human single nucleotide polymorphisms (SNPs) or insertion/deletion polymorphisms (indels) is 0.5, and the more deviation there is from 0.5, the more inclined it is to the presence of GI. So, based on this theory, we analyzed the AF of common germline-derived mutations, including SNPs and indels, in all samples of leukocytes, plasma ctDNA, and CSF ctDNA using the sequencing results of a panel covering 543 cancer-related genes to distinguish the presence of GI.

First, using peripheral blood leukocytes as a reference, the unfiltered germline mutation data were screened for shared mutations in the plasma ctDNA, CSF ctDNA, and tissues of all samples, and then mutations in leukocytes with a mutant allele frequency (AF) less than 0.2 or greater than 0.8 were filtered out to ensure that the mutations ultimately included in the analysis were germline-derived common mutations. The GI score for each sample was calculated by the formula:(1)$$GI\_Score=\frac{N_{GI\;mutations}}{N_{common\;mutations}}$$
which is equal to the proportion of mutations with an AF above the upper and lower quartiles (GI mutations) among the above mutations. 

### 2.7. Statistical Analysis

Statistics and Plots were generated by the R programming language 4.1.2. Pearson’s χ2 and Fisher’s exact tests were used to compare and analyze the clinicopathological characteristics of different GI score groups. After checking the assumption of normal distribution, the Wilcox test was used to assess the correlation between the GI score and sample type. Meningeal metastasis-free survival and overall survival were estimated by the Kaplan–Meier method. *p* < 0.05 was considered statistically significant.

## 3. Results

### 3.1. Cerebrospinal Fluid ctDNA Show More Genetic Variation Events Than Plasma ctDNA

We performed a comprehensive genomic profiling assay using a clinically validated and commercially available NGS target-sequencing panel covering 543 tumor-associated genes in order to compare the landscape of somatic mutations in CSF ctDNA and plasma ctDNA. Our results showed that the total number of mutations detected in CSF ctDNA (Figure 1A) was significantly higher than that in plasma ctDNA (Figure 1B,C, *p* < 0.001). Next, we compared single-nucleotide variants (SNVs) and copy number variants (CNVs) in CSF and plasma. The number of SNVs and the positive detection rate (PDR) in hotspot driver genes such as *EGFR* and *TP53* were significantly higher in CSF than in plasma (Figure 1D–F, *p* < 0.01). Consistently, a higher percentage of CNVs could be detected in CSF compared to that in plasma; 41 (73.2%) CNVs were detected in CSF and 7 (14%) CNVs were detected in plasma (Figure 1G, *p* < 0.001). CNVs of six driver genes were detected in both CSF and plasma among five MM patients, including *FGFR1, FGFR2, PIK3CA, ERBB2, RICTOR*, and *MDM2*. The vast majority had a higher gene copy number in CSF than in plasma (Figure 1H). In summary, our data suggest that MM-derived CSF ctDNA accumulate more genetic variants than plasma ctDNA.

### 3.2. A Panel-Developed GI Score Was Established to Describe the GI Status in CSF ctDNA and Verified Using Whole-Genome Sequencing-Based Copy Number Analysis

Based on the unfiltered mutation data-derived GI score, a high GI status was detected in CSF ctDNA in 37 of 56 (66.1%) MM patients, while genomic stability (GS) was seen in the remaining 19 (33.9%) patients (Table 2). The choice of a GI score cutoff value (about 0.07) was determined using the R package “cutoff”. Notably, unlike the large number of GI cases found in CSF ctDNA samples, GI was found in only three plasma cfDNA samples. Subsequently, considering that gene CNV reflects genomic stability to some extent, we added CNV analysis as a complimentary validation. Figure 2 shows the GI analysis results in five GI (A–E) and four GS (F–I) patients. 

Whole-genome sequencing (WGS) is the golden standard for determining GI. Therefore, to further validate the accuracy and reliability of our GI score, CSF from 8 of the 56 patients and paired plasma from 1 of them were collected for low-depth (10×) WGS analysis. Not surprisingly, and consistent with the panel-based sequencing analysis above, we found GI in the CSF of six MM patients (Figure 3A–E,H) and GS in two MM patients (Figure 3F,G). Notably, the CSF of patient numbered 48 had a high GI status, while his plasma had a GS status (Figure 3I).

### 3.3. Increased GI Was Detected in Mm Lesions-Derived CSF ctDNA Compared with Plasma ctDNA

We compared the GI score of all paired CSF ctDNA and plasma ctDNA samples. The results showed that the GI score was significantly higher in CSF (Figure 4A, *p* < 0.001). Moreover, as previously stated, the GI PDR was significantly higher in CSF (66.07%) than in plasma (6%), both in all 56 solid tumor MM samples (Figure 4B, *p* <0.001) and in the 45 LUAD MM samples with primary tumors as LUAD (Figure 4C, *p* < 0.001), suggesting that the GI status is higher in MM lesion-derived CSF ctDNA than it is in extracranial lesion-derived plasma ctDNA.

### 3.4. GI Is Associated with Poor Prognosis, High Intracranial Pressure, and Low Karnofsky Performance Status Scores in MM Patients

Compared with genomically stable MM patients, MM patients with a high GI status had worse clinical outcomes, including a shorter MM-free survival (MFS) and overall survival (OS) (MFS: median 1.7 vs. 2.9 years, *p* = 0.021; OS: median 2.5 vs 3.9 years, *p* = 0.038; Figure 5A,B). The Cox proportional hazards multivariate regression analysis revealed that GI was an independent risk factor for the development of MM in lung adenocarcinoma (risk ratio, MFS: 2.338; 95% CI, 1.109 to 4.929; *p* = 0.026) and an independent prognostic factor for OS (risk ratio, OS: 2.109; 95% CI, 1.030 to 4.317; *p* = 0.041, Table 3 and Table 4). In addition, the receiver operating characteristic curves for MFS and OS are shown in Appendix A.

Moreover, the intracranial pressure measured in MM patients with a high GI status was significantly higher than that in genomically stable MM patients, especially for the 45 patients with LUAD MM, whose intracranial pressure was subsequently revealed to be positively correlated with the GI score of the CSF (Figure 5C–F, *p* < 0.05). We also found that significantly more MM patients with a high GI status received a Lumboperitoneal (LP) shunt, which is only performed when the intracranial pressure exceeds 2.5 Kpa, suggesting that MM patients with a high GI status are more likely to develop severe intracranial hypertension with a concomitant increased risk of brain herniation (Figure 5G,H). 

In addition, the higher the Karnofsky Performance Status (KPS) score, the better the patient’s health and ability to tolerate treatment side effects. A low score results in many effective anti-tumor treatments not being implemented. We found that patients in the GI group showed low KPS scores, both in all 56 solid tumor MM patients and in all 45 LUAD MM patients (Figure 5I,J, *p* < 0.05).

### 3.5. Co-Mutations of Tp53 and Hotspot Driver Genes Were Associated with a High GI Score and Poor Prognosis 

Next, we compared the genetic characteristics of the GI and GS groups. The total number of mutations was significantly higher in the GI group than it was in the GS group, and further analysis revealed that the differences were mainly due to gene CNVs (Figure 6A–C). Moreover, differential gene mutation analysis showed that TP53 mutations occurred more frequently in MM patients with a high GI status (Figure 6D, *p* < 0.01) and were associated with a higher GI score (Figure 6E, *p* < 0.01). Meanwhile, we found that the mutation frequencies of *CDKN2A*, *APC*, *RICTOR*, *KMT2C*, *ERBB2*, *RB1*, *EGFR*, and other genes also tended to increase in the GI group (Appendix A, *p* < 0.2).

In addition, a comprehensive comparison of co-mutations between the two groups was performed. Multiple co-mutation events were identified in the GI group at a high frequency, such as *TP53/EGFR*, *TP53/RB1*, *TP53/ERBB2*, and *TP53/KMT2C* (Figure 6F–I, *p* < 0.05), and a higher GI score was detected in MM patients with the co-mutation of either *TP53/E**GFR*, *TP53/RB1*, *TP53/ERBB2*, or *TP53/KMT2C* (Figure 6J–K, *p* < 0.001), suggesting a close correlation between these co-mutations and a high GI status in MM. It is worth noting that the above co-mutations are correlated with worse clinical outcomes, including MSF and OS (Figure 6L–N, *p* < 0.05).

## 4. Discussion

While tumor biopsy is the standard for primary diagnosis, tissue biopsies are not available or sufficient for conducting a clinical molecular diagnosis to monitor disease progression or detect the relapse of MRD, especially in meningeal metastasis (MM) [21]. The genomic analysis of liquid biopsies, such as plasma, is currently a rapidly growing field because of its relative ease of collection and associated low cost [22]. It has been shown that the non-invasive characterization of solid tumor genomes can be achieved by detecting somatic mutations in circulating tumor DNA in plasma cfDNA [23,24]. However, tumor-derived cfDNA in the central nervous system is rarely detected in plasma due to the presence of the blood–brain barrier [25]. Therefore, a growing number of studies have investigated the sensitivity of CSF ctDNA for predicting MM carrying actionable mutations for targeted therapies [26,27]. In the present study, plasma and CSF were applied as surrogates of extracranial metastasis and MM, respectively, to compare the disparity in genomic profiles and GI status. Our data showed more genetic variation in CSF ctDNA compared to plasma ctDNA, suggesting the presence of GI in MM.

The presence of complex aneuploidies and polyploidies due to GI is observed in tumor types with a predisposition for metastasis, treatment resistance, and poor OS. These tumor types include triple-negative breast cancer, hepatobiliary cancer, and lung cancer [7,28,29]. However, there are no studies on GI in MM using CSF cfDNA. To the best of our knowledge, this study represents the first attempt to establish a GI score applicable to ctDNA and to explore the correlations between GI in CSF ctDNA and solid tumor MM prognosis. Several observations with implications for enhancing our understanding and the clinical relevance of GI in MM and the potential development of new therapies are evident from this study. We used an innovative method based on the analysis of unfiltered mutation data (which tend to be discarded) to demonstrate that GI, represented by GI scores calculated from CSF ctDNA, is an important independent prognostic biomarker for patients with MM. The shorter MFS in the GI group suggested that GI promotes the development of MM from LUAD. Notably, CSF ctDNA showed a high GI status in 66.1% of cases, whereas only 6% of plasma ctDNA samples showed a high GI status. Plasma ctDNA remains an excellent surrogate for the extracranial solid tumors in our analysis given that it shares a highly consistent GI score with the primary tumor tissue (Appendix A, r = 0.9, *p* < 0.01). As a result, the significantly higher GI scores and the higher proportion of samples with a high GI status in CSF actually reflect a difference between extracranial metastasis and MM, which could be explained by clonal evolution [30], that is, GI first occurred in the metastatic subclone of the primary tumor and then metastasized into the MM lesion to become the master clone before being detected. Another possibility is that GI occurs after MM.

Moreover, we also revealed that GI was associated with high intracranial pressure and low KPS scores. It has been demonstrated that MM-related intracranial hypertension, which was observed in about half of the patients with MM, leads to many dreaded neurological disorders, including headache, nausea, vomiting, and visual disturbances [31]. Patients with severe intracranial hypertension may develop brain herniation and even die. LP shunts have been shown to be effective in the treatment of MM-related severe intracranial hypertension and hydrocephalus [32,33]. Our data suggest that the proportion of patients who received an LP shunt—meaning they had very high intracranial pressure—was higher in the GI group than in the GS group. Several investigators have assessed the importance of the KPS score and found that a low KPS score predicts death in a relatively short period of time [34,35]. Therefore, the association of GI with these two clinical maladies suggests the need to determine the GI status in order to provide optimal therapeutic regimens in a timely manner and improve the prognosis of MM patients.

In addition, mutations in *TP53*, *CDKN2A*, *APC*, *RICTOR*, *KMT2C*, *ERBB2*, *RB1*, and *EGFR* appear to be associated with a high GI status of MM, as they have a high mutation frequency in the GI group, and patients with *TP53* mutations had higher GI scores. Notably, some co-mutations, such as *TP53/EGFR*, *TP53/RB1*, *TP53/ERBB2*, and *TP53/KMT2C*, also occurred more frequently in the GI group, which correlated with higher GI scores and a worse prognosis, although sole mutations did not have a significant impact on patient prognosis, which implied that GI-related distinct genetic mutation patterns affect the clinical outcome of solid tumor patients with MM and should be further investigated in the future.

The PARPs have emerged as promising targets for cancer therapy in recent decades due to their important role in DNA repair. PARP inhibitors have been approved by the U.S. Food and Drug Administration (FDA) for the treatment of breast, ovarian, fallopian tube, and prostate cancer patients. At the same time, the development of new PARP inhibitors with a new scope of application is in full swing [36,37,38]. Considering that cancers with GI caused by DNA repair defects or replication site emissions from oncogenes are particularly sensitive to PARP inhibitors [12], targeting GI alone or in combination with conventional therapy may be a promising treatment strategy for solid tumor patients with MM. Nevertheless, there are limitations to this study. This was a single-institution retrospective study with a small number of participants. All samples were archived specimens, and most were completely consumed at the time of panel sequencing, so our GI grouping results could not all be validated by WGS. In addition, previous reports have found that there appears to be a threshold for GI, with extremely unstable tumors having a better prognosis than generally unstable tumors [39]. Such extreme GI cases were not included in this study due to the limited sample size. Nevertheless, our study provides a relatively comprehensive look at GI in MM, reveals its impact on metastasis and prognosis, and is likely to improve future therapeutic approaches. Furthermore, a larger prospective clinical trial will be conducted, with the dynamic monitoring of GI before and during treatment in order to validate the prognostic value of the GI score and the therapeutic effects of PARP inhibitors in vitro and in vivo.

## 5. Conclusions

In conclusion, we established the first GI score based on CSF ctDNA germline-derived variant data, demonstrating the prognostic value of GI in MM. Our GI score is applicable to GI diagnosis—not only in liquid biopsies but also in tissue biopsies—and may be broadly available for multiple cancer types. We propose that targeting GI alone or in combination with conventional therapy for MM may be a promising treatment strategy.

## Figures and Tables

**Figure 1 cancers-14-05028-f001:**
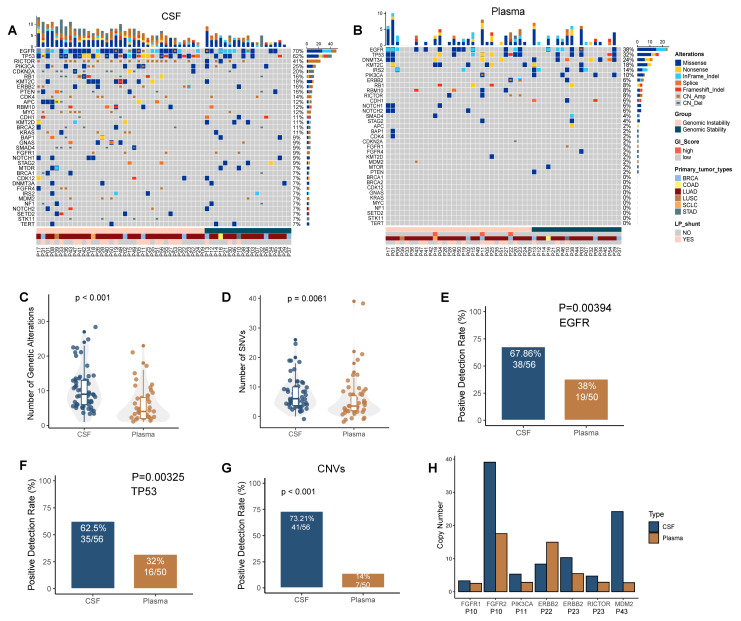
CSF ctDNA is more reflective of the genetic profile of meningeal metastasis than plasma ctDNA. (**A**,**B**) Oncoprint of CSF (**A**) and plasma (**B**). Each column represents a patient; each row represents a gene. Different types of mutations are colored differently. The top bar represents the total number of mutations that a patient has. A sidebar summarizes the frequency of mutations in this cohort. (**C**,**D**) The total number of genetic variants (**C**) and the total number of SNVs (**D**) in CSF and plasma. (**E**,**F**) The detection rates of EGFR mutations (**E**) and TP53 mutations (**F**). (**G**) The detection rate is defined as having any copy number variation detected from the panel. (**H**) Copy numbers of genes exhibiting CNV in both CSF and plasma.

**Figure 2 cancers-14-05028-f002:**
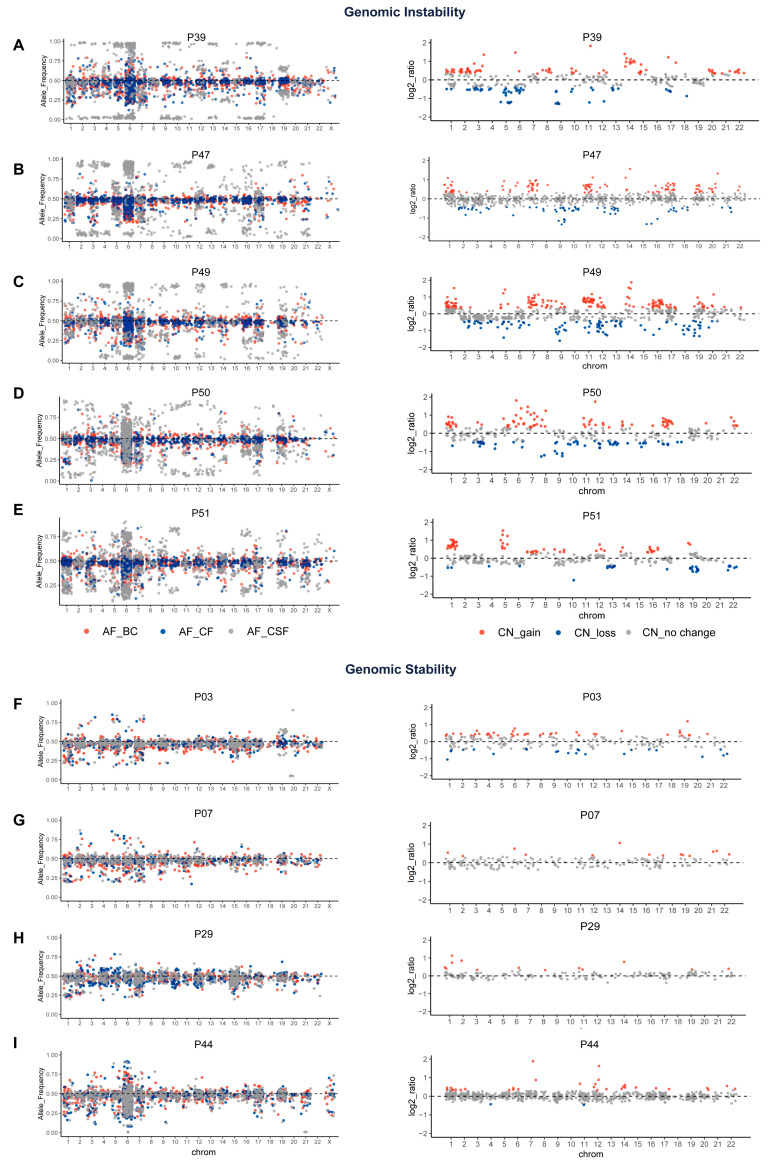
GS analysis of CSF and plasma based on a panel of 543 cancer-associated gene panels. The left side of the figure shows the allele frequency of shared mutations in leukocytes, plasma ctDNA and CSF ctDNA analyzed with unfiltered mutation data. The right side of the figure shows the distribution of the CSF gene copy number log2 ratio. All horizontal coordinates are chromosome locations. (**A**–**E**) Results of five patients with GI. (**F**–**I**) Results of four patients with GS. BC, white blood cells; CF, plasma; CSF, cerebrospinal fluid; CN, copy number.

**Figure 3 cancers-14-05028-f003:**
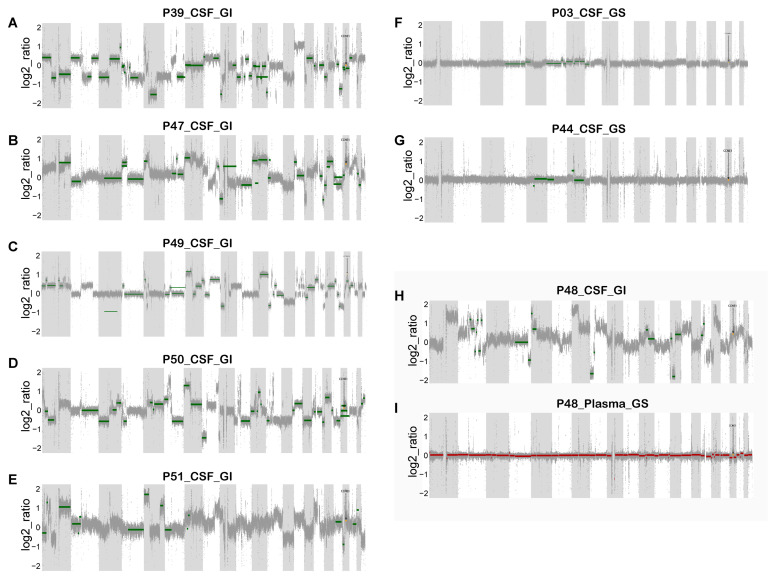
The reliability of the GI score was successfully verified by low-depth WGS copy number analysis in eight patients. (**A**–**E**,**H**) Changes in the gene copy number in the CSF of six patients with GI. (**F**,**G**) Changes in the gene copy number in the CSF of two patients with GS. (**I**) Changes in the gene copy number in the paired plasma of a genome-stable patient (P48).

**Figure 4 cancers-14-05028-f004:**
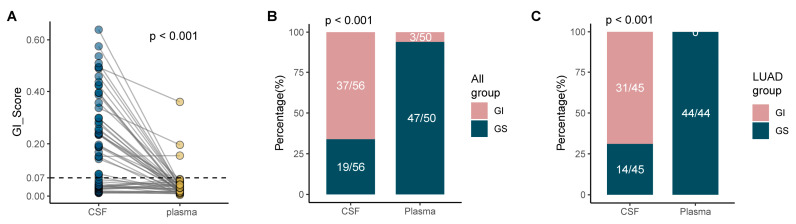
MM is associated with increased GI. (**A**) GI scores in the GI and GS groups. (**B**,**C**) The proportions of 56 solid tumor MM patients (**B**) and 45 LUAD MM patients (**C**) with GI in CSF and plasma ctDNA.

**Figure 5 cancers-14-05028-f005:**
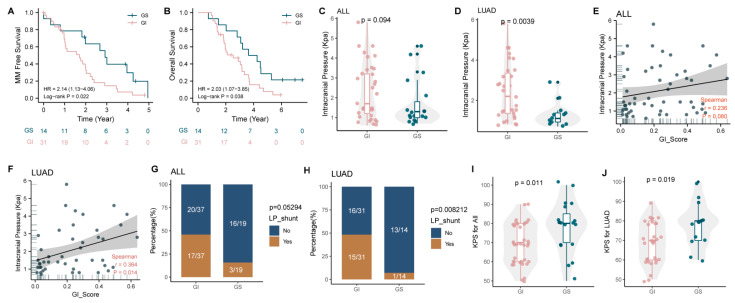
GI is associated with a poor prognosis, a low Karnofsky functional status score, and severe intracranial hypertension. (**A**,**B**) Analysis of MFS (**A**) and OS (**B**) in genomically unstable and genomically stable LUAD patients with MM. (**C**,**D**) Intracranial pressure in the GI and GS groups for all patients and (**D**) for 45 patients with LUAD (**C**). (**E**,**F**) Correlation analysis between intracranial pressure and GI scores. (**G**,**H**) The proportion of patients who have had LP shunts in the GI and GS groups. (**I**,**J**) KPS scores in the GI and GS groups. MM, meningeal metastasis; LUAD, lung adenocarcinoma; KPS, Karnofsky performance status; LP, Lumboperitoneal.

**Figure 6 cancers-14-05028-f006:**
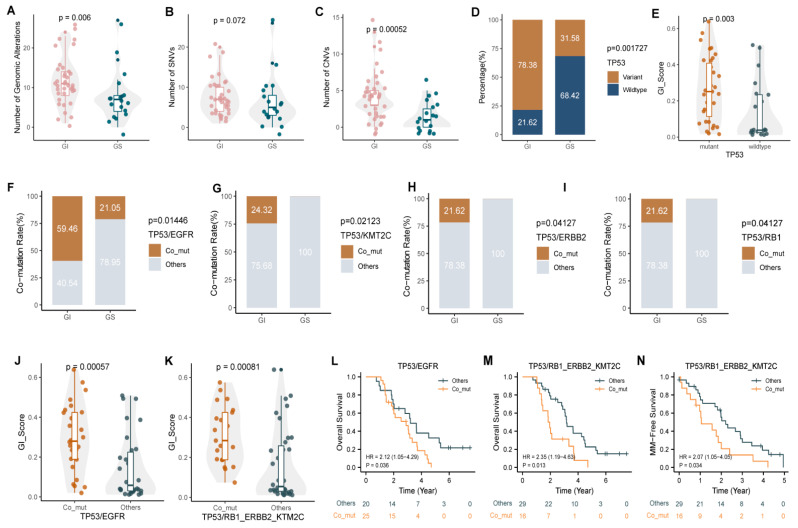
Comparison of gene characteristics between the GI and GS groups and their relationship with GI and prognosis. (**A**–**C**) The total number of gene mutations (**A**), the number of SNVs (**B**), and the number of CNVs (**C**) in the GI and GS groups. (**D**) Mutation rates of *TP53* in the GI and GS groups. (**E**) GI scores in CSF samples with/without *TP53* mutation. (**F**–**I**) Detection of co-mutations of *TP53*/*EGFR* (**F**), *TP53*/*KMT2C* (**G**), *TP53*/*ERBB2* (**H**), and *TP53*/*RB1* (**I**) in the GI and GS groups. (**J**,**K**) GI scores in CSF samples with/without corresponding *TP53* co-mutations. (**L**–**N**) The presence of *TP53* co-mutations was significantly associated with MM and poor prognosis.

**Table 1 cancers-14-05028-t001:** The basic clinical information of patients with meningeal metastases.

Characteristic	N = 56 ^1^
Gender	
Female	29 (52%)
Male	27 (48%)
Age	
≤55	27 (48%)
>55	29 (52%)
Primary tumor type	
BRCA	7 (12%)
COAD	1 (1.8%)
LUAD	45 (80%)
LUSC	1 (1.8%)
SCLC	1 (1.8%)
STAD	1 (1.8%)
KPS	70 (60, 80)
Intracranial pressure (Kpa)	1.5 (1.1, 3.1)
LP shunt	20 (36%)
Genomic status	
GI	37 (66.1%)
GS	19 (33.9%)

^1^ n (%); Median (IQR).

**Table 2 cancers-14-05028-t002:** Distributions of clinical parameters and GI scores in patients with different genomic statuses.

Characteristic	GI, N = 37 ^1^	GS, N = 19 ^1^	*p*-Value ^2^
Gender			0.074
Female	16 (43%)	13 (68%)	
Male	21 (57%)	6 (32%)	
Age			0.6
≤55	17 (46%)	10 (53%)	
>55	20 (54%)	9 (47%)	
Primary tumor type			0.4
BRCA	4 (11%)	3 (16%)	
COAD	0 (0%)	1 (5.3%)	
LUAD	31 (84%)	14 (74%)	
LUSC	1 (2.7%)	0 (0%)	
SCLC	1 (2.7%)	0 (0%)	
STAD	0 (0%)	1 (5.3%)	
Intracranial pressure (Kpa)	1.7 (1.2, 3.2)	1.3 (1, 1.8)	0.094
LP shunt			0.026
No	20 (54%)	16 (84%)	
Yes	17 (46%)	3 (16%)	
GI score	0.28 (0.19, 0.42)	0.03 (0.02, 0.04)	<0.001

^1^ n (%); Median (IQR). ^2^ Pearson’s Chi-squared test; Fisher’s exact test; Wilcoxon rank sum exact test.

**Table 3 cancers-14-05028-t003:** Univariate and multivariate Cox regression analyses of MFS.

Characteristics	Total (N)	Univariate Analysis		Multivariate Analysis	
		Hazard Ratio (95% CI)	*p* Value	Hazard Ratio (95% CI)	*p* Value
Sex	45				
Female	20	Reference			
Male	25	1.339 (0.688–2.606)	0.390		
Age	45				
≤55	21	Reference			
>55	24	0.992 (0.514–1.915)	0.981		
LP Shunt	45				
No	29	Reference			
Yes	16	1.669 (0.855–3.260)	0.133		
KPS	45	0.991 (0.967–1.016)	0.489		
Genomic Status	45				
GS	14	Reference			
GI	31	2.338 (1.109–4.929)	0.026	2.338 (1.109–4.929)	0.026

**Table 4 cancers-14-05028-t004:** Univariate and multivariate Cox regression analyses of OS.

Characteristics	Total (N)	Univariate Analysis		Multivariate Analysis	
		Hazard Ratio (95% CI)	*p* Value	Hazard Ratio (95% CI)	*p* Value
Gender	45				
Female	20	Reference			
Male	25	1.764 (0.915–3.400)	0.090		
Age	45				
≤55	21	Reference			
>55	24	0.872 (0.459–1.658)	0.677		
LP Shunt	45				
No	29	Reference			
Yes	16	1.473 (0.764–2.840)	0.248		
KPS	45	0.978 (0.953–1.003)	0.082		
Genomic Status	45				
GS	14	Reference			
GI	31	2.109 (1.030–4.317)	0.041	2.109 (1.030–4.317)	0.041

## Data Availability

The data presented in this study are available within the article or Appendix A.

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
