# Peer review of "Genomic Instability in Cerebrospinal Fluid Cell-Free DNA Predicts Poor Prognosis in Solid Tumor Patients with Meningeal Metastasis"

_cancers, 2022, doi:10.3390/cancers14205028_

Round 1
Reviewer 1 Report
In the article entitled “Genomic instability in cerebrospinal fluid cell-free DNA predicts poor prognosis in solid tumor patients with meningeal metastases” Wang P. et al performed a retrospective study by analysing cell free DNA isolated by cerebrospinal fluid of 56 patients affected by different primary tumours with meningeal metastasis.
The study is interesting, and the results might be useful for setting novel target therapies against solid tumours with meningeal metastasis.
Minor points:
1) With ctDNA you mean circulating tumor DNA? Please explain the abbreviation the first time it appears in the text
2) Panel (A) and (B) of Figure 1 are not mentioned in the manuscript, please add the citation in the paragraph 3.1
3) Line 259: TP53/ERFR
Author Response
-
- With ctDNA you mean circulating tumor DNA? Please explain the abbreviation the first time it appears in the text
Response:
We appreciate the reviewer’s constructive comments and suggestions. The ctDNA mentioned in the text is circulating tumor DNA, and we have now included the explanation of ctDNA when it first appears in the text as suggested by the reviewers. (Page 1, Line 16).
- Panel (A) and (B) of Figure 1 are not mentioned in the manuscript, please add the citation in the paragraph 3.1
Response:
We thank the reviewer’s comment and have supplemented the citation of panels (A) and (B) of Figure 1 in paragraph 3.1 of the revised manuscript (Page 4, Line 158).
- Line 259: TP53/ERFR
Response:
Sorry for the mistake, we have revised it in the revision (Page 11, Line 277).
Reviewer 2 Report
This study is very timely and addresses a current unresolved issue in solid tumours with meningeal metastasis (MM). The authors investigate the levels of genomic instability in a cohort of 56 patients and examine both plasma and cerebral fluid for cfDNA and its level of genomic instability. They detect a higher level of instability in the cerebral fluid. The patients present with high or low levels of genomic instability, which is reflected in their overall progression free survival and rate of progression as well as in their KPS scores.
Overall, this study is very solid and has important implications for the clinic; the methods are well described.
Minor: please proof-read the English.
Author Response
We appreciate the reviewer’s constructive comment. The revision has been edited by a native English speaker to prevent any typographical and grammatical errors.
We have uploaded the proof of the English editing along with the revision for your reference.